# A Vision Foundation Model for Cataract Surgery Using Joint-Embedding Predictive Architecture

**Nisarg A. Shah**[*][1]                                            SNISARG812@GMAIL.COM
[1] *Johns Hopkins University, Baltimore, USA*

**Mingze Xia**[*][1]                                               MXIA8@JHU.EDU
**Subhasri Vijay**[1]                                             SVIJAY2@JHU.EDU
**Shameema Sikder**[2,3]                                          SSIKDER1@JHMI.EDU
[2] *Malone Center for Engineering in Healthcare, Baltimore, USA*
[3] *Wilmer Eye Institute, Johns Hopkins University, Baltimore, USA*

**S. Swaroop Vedula**[2]                                          SWAROOP@JHU.EDU
**Vishal M. Patel**[1]                                            VPATEL36@JHU.EDU

**Editors:** Accepted for publication at MIDL 2025

## Abstract

Vision foundation models can automate surgical video analysis, enabling applications that support patient care and training. For cataract surgery, existing models are limited by small datasets, privacy concerns, and poor generalizability. In this paper, we introduce JHU-VPT(JEPA), a self-supervised vision foundation model leveraging Joint-Embedding Predictive Architecture (JEPA) to learn spatiotemporal representations via latent feature prediction on a large corpus of unlabeled cataract videos, without requiring extensive labeled datasets or pixel-level reconstruction. JHU-VPT(JEPA) is pretrained on 2591 videos from multiple sites, capturing diverse surgical techniques and styles. Comprehensive evaluations on step recognition, surgical feedback, and skill assessment tasks show JHU-VPT(JEPA) outperforms existing methods. Its effectiveness is evident even with attentive probing using a frozen encoder, highlighting feature robustness and addressing privacy by not needing raw video access for downstream tasks. Our approach offers a scalable, generalizable, and privacy-preserving solution for surgical video analysis, with significant potential to advance patient care and surgical education.

**Keywords:** Surgical Pretraining, Joint Embedding Predictive Network, Cataract Surgery

## 1. Introduction

Vision foundation models analyzing surgical videos can substantially impact global patient care. Intraoperative surgical videos offer rich data for algorithms enabling critical applications like activity recognition, and skill/feedback prediction for surgeon learning and evaluation (Maier-Hein et al., 2017; Yu et al., 2019; Padoy, 2019; Shah et al., 2025b). Surgical data science has accelerated models for analyzing surgical videos. However, current models face constraints like small/convenience datasets (Shah et al., 2023), limited evaluation, and poor generalizability (Lecuyer et al., 2020; Padoy, 2019; Funke et al., 2019). While foundation models are rapidly being trained for applications in several domains (Kang et al., 2023; Lu et al., 2023), vision foundation models pretrained on large surgical video datasets have not yet been developed.

---

[*] Contributed equally

Self-supervised learning (SSL) has emerged as a powerful paradigm to leverage large corpora of unlabeled video data and train vision foundation models. Traditional SSL methods for medical imaging often involve multimodal cues, e.g., textual radiology reports paired with X-ray images (Boecking et al., 2022; Moon et al., 2022). By contrast, surgical videos typically lack accompanying text annotations, necessitating visual self-supervised schemes. To address this, we propose a self-supervised approach for complex spatio-temporal surgical video information. Building on the Joint-Embedding Predictive Architecture (JEPA) (Assran et al., 2023; Weimann and Conrad, 2024), our method focuses on *feature prediction* in latent space, a strategy that captures both spatio-temporal coherence and surgical scene semantics without requiring direct pixel-level reconstruction.

Unlike prior self-supervised strategies that primarily rely on contrastive learning or masked autoencoders (MAEs) (He et al., 2022; Tong et al., 2022; Shah et al., 2025a), our approach masks tokens and predicts them in a latent feature space rather than reconstructing raw pixel values. This reduces the effect of low-level artifacts such as reflections and blur, while emphasizing higher-level semantic details that are crucial for tasks like recognition, feedback, and skill assessment. By eliminating the need for a pixel decoder and a separate reconstruction loss, the learning process is simplified to focus solely on meaningful feature extraction. Moreover, the use of an exponential moving average (EMA) stabilizes training by reducing gradient noise, which is particularly important in surgical video analysis where lighting changes and rapid motion clips can disrupt learning. We develop our model, *JHU-VPT(JEPA):Cataract*, which we refer to as JHU-VPT(JEPA), by pretraining on a large corpus of cataract surgery videos including multiple sites and surgeons. The dataset diversity allows the learning of domain-robust embeddings. The resultant representations can be shared more readily than raw videos (protecting patient privacy), and they excel in label-scarce scenarios, reducing the need for extensive manual annotations and data-hungry fine-tuning protocols. We comprehensively evaluate JHU-VPT(JEPA)'s learned embeddings on three key tasks: (1) **Step Recognition**, wherein the aim is to identify surgical steps or phases; (2) **Surgical Feedback**, to predict specific performance feedback for the surgeon; and (3) **Skill Assessment**, which is essential for both surgeon training and credentialing. By varying the size of the annotated subsets used for fine-tuning, we show that JHU-VPT(JEPA) achieves strong performance even with limited labels, highlighting its data efficiency. Furthermore, we validate cross-domain generalization by testing on previously unseen videos, demonstrating JHU-VPT(JEPA)'s capacity to adapt to new surgical styles or camera configurations.

**Contributions.** In summary, our main contributions are:

- **JEPA-based approach for cataract videos.** We introduce JHU-VPT(JEPA), a novel architecture for surgical video analysis employing latent space feature prediction to learn rich spatio-temporal representations. These are validated via attentive probing with a frozen encoder, confirming high transferability and effectiveness without fine-tuning.

- **Large-scale video pretraining using a large dataset.** We use an extensive, multi-institution unlabeled surgical video dataset, ensuring robust, domain-generalizable embeddings.

- **Comprehensive downstream evaluation.** We test JHU-VPT(JEPA) on three important tasks—step recognition, surgical feedback, and skill assessment—showing notable gains with varying labeled data, underscoring its potential clinical utility.

## 2. JHU-VPT(JEPA): Cataract model

In this section, we describe our proposed *Vision Foundation Model for Cataract Surgery* **JHU-VPT(JEPA)**, which builds upon the JEPA principle (Garrido et al., 2024; Bardes et al., 2024) for learning rich, robust visual representations from cataract surgery videos. Our goal is to exploit *feature prediction* as a stand-alone objective, enabling the model to learn meaningful spatio-temporal embeddings without extra supervision. A high-level overview of JHU-VPT(JEPA) is shown in Figure 1.

### 2.1. Overview

At the core of feature prediction as a stand-alone objective, the model learns by predicting the representation of a target input $\mathbf{y}$ from the representation of a context input $\mathbf{x}$. Specifically, an encoder $E_\psi(\cdot)$ projects $\mathbf{x}$ into latent space, while a predictor $P_\phi(\cdot)$ attempts to recover the embedding of $\mathbf{y}$ given $\mathbf{x}$. A conditioning variable $\delta$, indicating the transformation or corruption that links $\mathbf{x}$ and $\mathbf{y}$, guides the predictor to generate distinct outputs for different transformations. In our setting, $\mathbf{x}$ and $\mathbf{y}$ are disjoint spatio-temporal patches from a surgical clip, and $\delta$ encodes the masking pattern (or offset) between these two regions.

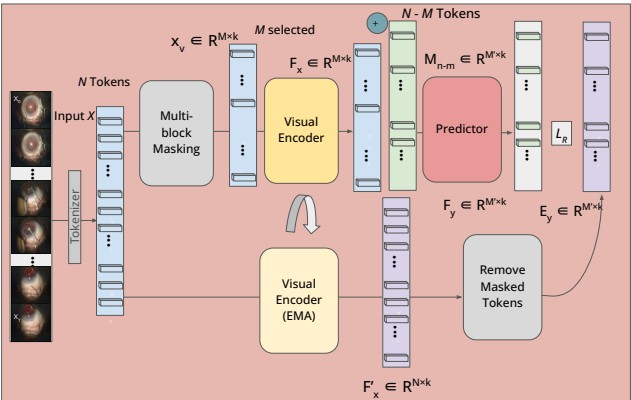

Figure 1: Overview of the JHU-VPT(JEPA) architecture. The framework consists of Block Masking, an Encoder, a Predictor, and an EMA-updated Target Encoder. The Encoder processes the non-masked tokens, predicting their feature representations. The Predictor combines these representations with learnable mask tokens and a conditioning variable to predict the embeddings of masked regions. The Target Encoder encodes all tokens, generating target embeddings for the feature-prediction loss.

## 2.2. Training Objective

To learn robust representations, we train the visual encoder $E_\psi(\cdot)$ and the predictor $P_\phi(\cdot)$ via a feature-prediction loss. Let the *context* region $\mathbf{x}$ and *target* region $\mathbf{y}$ be two non-overlapping subsets of video tokens from a video $\mathbf{X}$, selected according to a masking scheme (see Section 2.3). We define the loss function to encourage the predicted representation of $\mathbf{y}$ to match the actual representation of $\mathbf{y}$, generated by a *target encoder* $E_{\bar{\psi}}(\cdot)$. Concretely, we minimize:

$$\min_{\psi, \phi} \; \left\| P_\phi\big(E_\psi(\mathbf{x}), \delta\big) \; - \; \mathrm{sg}\big(E_{\bar{\psi}}(\mathbf{y})\big) \right\|_1, \tag{1}$$

where $\mathrm{sg}(\cdot)$ is a stop-gradient blocking updates to $E_{\bar{\psi}}(\cdot)$. In practice, $\bar{\psi}$ is maintained as an exponential moving average (EMA) of $\psi$, consistent with prior work that mitigates representation collapse (Garrido et al., 2024). Using L1 loss and a stop-gradient on the target encoder prevents trivial solutions (i.e., feature collapse) by forcing the encoder and predictor to capture meaningful spatio-temporal information in the surgical video.

**Collapse Prevention.** Combining an EMA target encoder, a stop-gradient, and a predictor prevents representation collapse in various self-supervised contexts (Grill et al., 2020; Assran et al., 2023). Intuitively, $\bar{\psi}$ changes more slowly than $\psi$, compelling $E_\psi(\mathbf{x})$ to capture detailed information needed by $P_\phi(\cdot)$ to match the slowly evolving target representation. This strategy drives the encoder to encode distinct semantic cues (e.g., instruments, ocular structures, movements) rather than collapsing to constant outputs.

## 2.3. Prediction Task and Masking Strategy

We implement the feature-prediction objective using a masked modeling approach. Each video clip is partitioned into 3D tokens, and large continuous blocks are sampled to form the masked regions $\mathbf{y}$; the remaining tokens constitute the visible regions $\mathbf{x}$. Applying large or continuous masks across time creates a challenging prediction task, encouraging the model to capture dynamic interactions between surgical instruments and ocular tissue.

To achieve this, we use multi-block masking (Bardes et al., 2024). First, short-range masks involve sampling several small blocks (e.g., 8) that cover about 15% of each frame, applied consistently across all frames. This forces the model to rely on temporal cues to infer fine-grained details and quick instrument movements. Second, long-range masks involve sampling fewer, larger blocks (e.g., 2) covering approximately 70% of each frame and extending over time, forcing the model to understand broader surgical phases and slower eye changes from limited visible areas. This multi-block masking strategy challenges the predictor to reconstruct features of large masked regions from small visible segments, enhancing the model's understanding of actions and anatomy in surgery videos.

## 2.4. Implementation Details

JHU-VPT(JEPA) comprises three learnable modules and an EMA-updated target encoder. **Tokenizer:** The tokenizer converts the raw video $\mathbf{X} \in \mathbb{R}^{T \times C \times H \times W}$ into non-overlapping 3D tokens representing spatio-temporal volumes. We apply a 3D convolutional layer with kernel and stride $(t, h, w)$, producing $N = \frac{T}{t} \times \frac{H}{h} \times \frac{W}{w}$ tokens, each of dimension $k$. Fixed 3D

positional encodings (He et al., 2022; Tong et al., 2022) are added to retain spatio-temporal information.

**Encoder** $E_\psi(\cdot)$: The encoder is a Vision Transformer (ViT) backbone (Dosovitskiy et al.; Arnab et al., 2021) that processes the visible tokens $\mathbf{x}$, producing an embedding $\mathbf{F}_x \in \mathbb{R}^{|\mathbf{x}| \times d}$, where $d$ is the embedding dimension. This embedding is passed to the predictor.

**Predictor** $P_\phi(\cdot)$: The predictor is a lightweight transformer that maps $\mathbf{F}_x$ to a predicted embedding $\widetilde{\mathbf{F}}_y$. It also receives learnable mask tokens $\mathbf{M}$ (one per masked patch) with positional encodings and the conditioning variable $\delta$, which encodes positional offsets or transformations between $\mathbf{x}$ and $\mathbf{y}$. Formally,

$$\widetilde{\mathbf{F}}_y = P_\phi\big(\mathbf{F}_x, \mathbf{M}, \delta\big). \tag{2}$$

**Target Encoder** $E_{\bar{\psi}}(\cdot)$: The target encoder is an EMA copy of the encoder, updated at each training iteration by

$$\bar{\psi} \leftarrow \alpha\, \bar{\psi} + (1 - \alpha)\, \psi, \tag{3}$$

where $\alpha \in [0, 1)$ is a momentum coefficient. It processes the masked tokens $\mathbf{y}$, generating $\mathbf{F}_y$ for the loss in Eq. (1).

## 2.5. Pretraining Architecture Analysis

Predicting large, masked video region representations from limited visible cues allows JHU-VPT(JEPA) to capture fine-grained details and long-range context in cataract surgery. The joint-embedding mechanism directs the encoder to focus on discriminative aspects like surgical instruments, subtle eye movements, and relevant clinical steps. The EMA target encoder, stop-gradient, and predictor prevent collapse, enabling learning of temporally coherent, anatomically relevant features. This design is scalable to various downstream tasks, including surgical phase recognition and skill assessment, and demonstrates strong generalization with minimal labeled data. Section 3.3 demonstrates JHU-VPT(JEPA)'s effectiveness in capturing real-world surgical workflow complexities while maintaining low annotation requirements.

## 2.6. Downstream Task Evaluation

After pretraining JHU-VPT(JEPA), we evaluate its representations on downstream tasks using two approaches: *fine-tuning* and *attentive probing*. In fine-tuning, we initialize the encoder $E_\psi(\cdot)$ with the pretrained weights and attach a linear classification head. The entire model, including the encoder and the classification head, is then optimized jointly on the downstream dataset.

In contrast, attentive probing keeps the pretrained encoder $E_{\bar{\psi}}(\cdot)$ fixed to assess the quality of the learned features without updating them. We introduce a learnable cross-attention layer with a query token that attends to the output features of the frozen encoder. The output of the cross-attention layer is added to the query token via a residual connection and passed through a two-layer multilayer perceptron (MLP) for prediction:

$$\mathbf{h} = \mathrm{MLP}\big(\mathbf{q} + \mathrm{CrossAttn}(\mathbf{q}, E_{\bar{\psi}}(\mathbf{x}))\big), \tag{4}$$

where **q** is the learnable query token, and **h** is the output used for classification or regression tasks. Attentive probing evaluates the robustness of the pretrained features while keeping the feature extractor unchanged, ensuring that the representation quality is not influenced by further training. This approach is useful when labeled data is limited or when data privacy restrictions prevent sharing raw videos, as it allows training downstream models on new tasks using shared features without accessing the raw video data.

## 2.7. Datasets

For **Pretraining** JHU-VPT(JEPA), we assembled a multi-institutional dataset of 2,591 unlabeled cataract surgery videos. This dataset includes 1,838 internal videos (avg. 30 min, 59 fps) and 753 from Cataract-1k (Ghamsarian et al., 2024) (avg. 8 min), totaling 2591 unique videos. We did not pretrain on the Cataract-1k videos for which step recognition annotations are available. All videos were subsampled to 1 *fps* and resized to $250 \times 250$ pixels for pretraining, following prior protocols (Gao et al., 2021; Twinanda et al., 2016). CSMAE (Shah et al., 2025a) was pretrained on the D-450 dataset (an extension of D99 videos), following the methodology for MAE-based pretraining (Bandara et al., 2023).

We evaluated JHU-VPT(JEPA) on three downstream tasks: step recognition, surgical feedback, and skill assessment. For **step recognition**, experiments were conducted under both low-data (10%, 25%, 50%) and full-data settings using four cataract surgery datasets: Cataract-101 (Schoeffmann et al., 2018), D99 (Yu et al., 2019), Cataract-1k (subset for which annotations were provided with the original dataset) (Ghamsarian et al., 2024), and a larger subset of Cataract-1k which we internally annotated (referred to as Cataract-1k-JHU and includes the annotated videos in the original dataset). Cataract-101 contains 101 videos at 25 *fps* with 10 annotated steps and a resolution of $720 \times 540$ pixels, split into 50 training, 10 validation, and 40 testing videos (Shah et al., 2023). D99 comprises 99 videos at 59 *fps* with 12 annotated steps and a resolution of $640 \times 480$ pixels, partitioned into 60 training, 20 validation, and 19 testing videos (Shah et al., 2023). For Cataract-1k, we used 25 training, 7 validation, and 24 testing videos. For Cataract-1k-JHU, we employed 181 training, 31 validation, and 91 testing videos. All evaluation videos were subsampled to 1 *fps* and resized to $250 \times 250$ pixels for consistency.

Table 1: Qualitative Results on Step Recognition Task for Attentive and Linear Probing Methods across four Datasets. JHU-VPT(JEPA) consistently outperforms Video-MAE across both probes when pretrained on the same pretraining set (D-2591). Results are presented as percentage accuracies, with improvements of our method relative to VideoMAE shown in green.

| Dataset | Attentive Probing | | Linear Probing | |
|---|---|---|---|---|
| | VideoMAE | Ours | VideoMAE | Ours |
| Cataract-101 | 79.31 | 89.82 (+13.3%) | 65.49 | 67.04 (+2.4%) |
| Cataract-1k | 63.75 | 79.58 (+24.8%) | 52.26 | 52.51 (+0.5%) |
| Cataract-1k-JHU | 70.03 | 83.65 (+19.5%) | 58.76 | 59.04 (+0.5%) |
| D99 | 66.13 | 77.20 (+16.7%) | 47.75 | 48.75 (+2.1%) |

In the **surgical feedback** task, we evaluated JHU-VPT(JEPA) on feedback items (Xia et al., 2025) during the capsulorhexis step using the D99 dataset (Hira et al., 2022) of 99

videos. Frames were resized to $224 \times 224$ pixels, applying data augmentations like rotation and color jitter. Data was split into training (60%), validation (20%), and testing (20%) sets; we repeated experiments with three random splits and averaged the results.

For **skill assessment** in the main incision and capsulorhexis steps, we used 56 videos from D99 and an additional 37 videos captured under consistent conditions. Expert surgeons evaluated the videos using ICO-OSCAR:Phacoemulsification (Puri et al., 2017). Skill was categorized as novice (scores 2–4) and expert (score 5) for main incision, and similarly for capsulorhexis, following (Hira et al., 2022; Kim et al., 2019).

## 3. Experiments and Results

### 3.1. Evaluation Metrics

We evaluate JHU-VPT (JEPA) using Accuracy, Precision, Recall, and Jaccard Index for step recognition (Shah et al., 2023; Kim et al., 2019; Shah et al., 2025b), and Accuracy, Sensitivity, Specificity, and AUC for surgical feedback and skill assessment.

### 3.2. Comparison to State-of-the-Art Cataract Pretraining Models

We compare JHU-VPT(JEPA) with existing pretraining models on the Cataract-101, D99, Cataract-1k, and Cataract-1k-JHU datasets. Table 1 shows the model's performance in attentive and linear probing settings (frozen encoder). Our approach consistently outperforms VideoMAE in both, especially with attentive probing (over 20% improvement for Cataract-1k), highlighting pretrained feature effectiveness and applicability in privacy-aware training. Table 3 shows that in attentive probing, JHU-VPT (JEPA) consistently outperforms Video-MAE (Tong et al., 2022) (pretrained on the same set) across all four datasets, often by over 10 percentage points with full data. Even with only 10% labeled data on Cataract-1k-JHU, JHU-VPT (JEPA) achieves 63.81% accuracy versus VideoMAE's 58.36%. Strong gains on Cataract-101 and D99 (different sources) indicate better generalization, showing JHU-VPT (JEPA)'s learned features effectively capture key surgical patterns without needing encoder updates in downstream tasks.

In full fine-tuning (Tab. 2, Fig. 3), JHU-VPT(JEPA) is competitive with CSMAE (Shah et al., 2025a) across several metrics/datasets. Pretraining on larger, diverse data enhances JHU-VPT(JEPA)'s performance, emphasizing data diversity's role in SSL for surgical video. While JHU-VPT(MAE) (VideoMAE on our pretraining set) often yields higher accuracy with full fine-tuning (Feichtenhofer et al., 2022), JHU-VPT(JEPA)'s predictive objective learns abstract, high-level features that excel with attention probes (frozen encoder), showcasing robustness. In contrast, full fine-tuning adjusts all parameters, which can perturb these representations. Overall, JHU-VPT(JEPA)'s feature prediction with large, diverse pretraining data yields significant gains in cataract surgery analysis, and its robust features enable privacy-constrained surgical video analysis with minimal fine-tuning (Garrido et al., 2024).

Overall, JHU-VPT(JEPA)'s feature prediction approach, enabled by large and diverse pretraining data, yields significant performance gains in cataract surgery analysis. Its robust features allow surgical video analysis in privacy-constrained scenarios with minimal fine-tuning (Garrido et al., 2024).

Table 2: Quantitative results of step recognition from different methods on the Cataract-101 and D99 datasets.

| Method | Cataract-101 | | | | D99 | | | |
|---|---|---|---|---|---|---|---|---|
| | Jaccard | Precision | Recall | Accuracy | Jaccard | Precision | Recall | Accuracy |
| ResNet(He et al., 2016) | 62.58 | 76.68 | 74.73 | 82.64 | 37.98 | 54.76 | 52.28 | 72.06 |
| SV-RCNet(Jin et al., 2017) | 66.51 | 84.96 | 76.61 | 86.13 | 39.15 | 58.18 | 54.25 | 73.39 |
| OHFM(Yi and Jiang, 2019) | 69.01 | 85.37 | 78.29 | 87.82 | 40.01 | 59.12 | 55.49 | 73.82 |
| TeCNO(Czempiel et al., 2020) | 70.18 | 86.03 | 79.52 | 88.26 | 41.31 | 61.56 | 55.81 | 74.07 |
| TMRNet(Jin et al., 2021) | 71.83 | 85.09 | 82.44 | 89.68 | 41.42 | 61.37 | 56.02 | 75.11 |
| Trans-SVNet(Gao et al., 2021) | 72.32 | 86.72 | 81.12 | 89.45 | 42.06 | 60.12 | 56.36 | 74.89 |
| ViT(Dosovitskiy et al.) | 64.77 | 78.51 | 75.62 | 84.56 | 38.18 | 55.15 | 53.60 | 72.45 |
| TimesFormer(Bertasius et al., 2021) | 75.97 | 85.38 | 84.47 | 90.76 | 42.69 | 64.24 | 55.17 | 77.83 |
| STMAE(Feichtenhofer et al., 2022) | 70.54 | 81.47 | 78.67 | 85.29 | 41.67 | 59.38 | 53.22 | 74.16 |
| VideoMAE(Tong et al., 2022) | 71.39 | 82.13 | 80.16 | 86.47 | 42.58 | 61.24 | 56.35 | 74.39 |
| CSMAE(Shah et al., 2025a) | 76.82 | 84.26 | 86.73 | 89.83 | 43.51 | 64.32 | 52.45 | 78.14 |
| JHU-VPT(MAE) | **79.95** | 87.80 | **89.10** | **92.00** | **49.95** | **64.78** | **64.46** | **78.69** |
| JHU-VPT(JEPA) | 79.58 | **87.88** | 88.89 | 91.52 | 43.63 | 55.39 | 62.19 | 75.61 |

Table 3: Comparison of Step Recognition Accuracy across different dataset splits (100%, 50%, 25%, 10%) with Attentive Probing. Results are presented as percentage accuracies. For each split, the higher value between VideoMAE (Tong et al., 2022) and JHU-VPT(JEPA) - Ours is highlighted in bold.

| Dataset | 100% Split | | 50% Split | | 25% Split | | 10% Split | |
|---|---|---|---|---|---|---|---|---|
| | VideoMAE | Ours | VideoMAE | Ours | VideoMAE | Ours | VideoMAE | Ours |
| Cataract-101 | 79.31 | **89.82** | 72.41 | **84.79** | 70.64 | **79.73** | **58.60** | 56.95 |
| D99 | 66.13 | **77.20** | 51.10 | **71.51** | 47.56 | **63.21** | 42.10 | **45.56** |
| Cataract-1k | 63.75 | **79.58** | **59.65** | 58.80 | **46.91** | 45.09 | **36.61** | 35.12 |
| Cataract-1k-JHU | 70.03 | **83.65** | 66.18 | **80.71** | 63.93 | **74.55** | 58.36 | **63.81** |

**Qualitative Results:** As shown in Fig. 2, our JHU-VPT (JEPA) model outperforms VideoMAE in overall prediction quality across all four datasets. We observe clearer and more accurate step-transition boundaries, as well as reduced temporal jitter, which indicate more robust and stable features from JHU-VPT (JEPA) for Attentive Probing. These improvements arise from the JEPA-based latent space modeling, which emphasizes richer spatiotemporal representations over pixel-level reconstructions.

### 3.3. Comparison on Feedback and Skill Performance

Table 4(a) shows that our method improves feedback prediction by approximately 13% in AUC compared to TimeSformer (Bertasius et al., 2021). Compared to other methods, JHU-VPT(JEPA) improves specificity, i.e., reduces false positives, indicating that the model has meaningful discrimination between positive and negative labels. Table 4(b,c) demonstrates our model's performance on skill assessment for main incision and capsulorhexis. We observe a steady improvement of 10%-20% for both phases, which highlights the robustness of its learned feature representations across various phases.

## 4. Conclusion

We introduced JHU-VPT(JEPA) for cataract surgery video analysis. JHU-VPT(JEPA), using latent feature prediction, captures rich spatio-temporal representations without needing

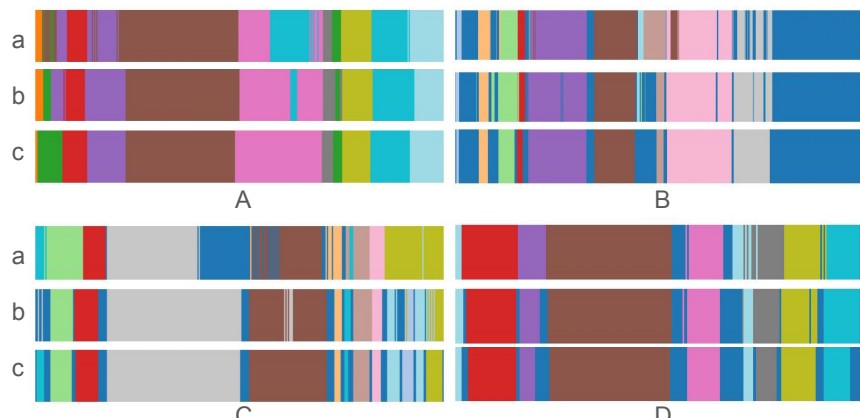

Figure 2: Qualitative results for the step recognition task. Subfigures A, B, C, and D correspond to the datasets Cataract-101, D99, Cataract-1k, and Cataract-1k-JHU, respectively. Within each dataset, the small subfigures a, b, and c are ribbon plots showing the prediction for VideoMAE (Tong et al., 2022), JHU-VPT (JEPA) trained with Attentive Probing, and the ground truth, respectively.

Table 4: Model evaluation for predicting feedback items, skill assessment in main incision, and skill assessment in capsulorhexis.

| Model | Accuracy | Sensitivity | Specificity | AUC |
|---|---|---|---|---|
| **Feedback Prediction (Table 4a)** | | | | |
| CNN-LSTM (Wan et al., 2024) | $76.3 \pm 1.6$ | $\mathbf{94.3 \pm 1.5}$ | $15.3 \pm 1.7$ | $0.659 \pm 0.049$ |
| CNN-LSTM-GNN (Xia et al., 2025) | $75.0 \pm 1.1$ | $85.6 \pm 2.6$ | $34.5 \pm 6.7$ | $0.559 \pm 0.048$ |
| JHU-VPT(MAE) (D-2591) | $80.4 \pm 2.9$ | $93.1 \pm 8.0$ | $35.9 \pm 8.2$ | $0.817 \pm 0.032$ |
| TimeSformer (Bertasius et al., 2021) | $77.2 \pm 1.1$ | $85.7 \pm 5.9$ | $40.2 \pm 12.4$ | $0.710 \pm 0.066$ |
| JHU-VPT(JEPA) (Ours) | $\mathbf{82.3 \pm 1.4}$ | $92.6 \pm 6.9$ | $\mathbf{40.8 \pm 16.5}$ | $\mathbf{0.842 \pm 0.045}$ |
| **Main Incision Skill Assessment (Table 4b)** | | | | |
| CNN-LSTM (Hira et al., 2022) | 63.0 | 92.0 | 36.0 | 0.64 |
| ViT (Dosovitskiy et al.) | 62.0 | 10.0 | 100.0 | 0.55 |
| JHU-VPT(JEPA) (Ours) | 73.0 | 60.0 | 63.0 | 0.72 |
| **Capsulorhexis Skill Assessment (Table 4c)** | | | | |
| ResNet-101 (He et al., 2016) | 62.0 | 76.0 | 80.0 | 0.45 |
| STMAE (Feichtenhofer et al., 2022) | 66.0 | 68.0 | 80.0 | 0.55 |
| JHU-VPT(MAE) (D-2591) | 71.0 | 85.0 | 90.0 | 0.55 |
| JHU-VPT(JEPA) (Ours) | 80.0 | 70.0 | 56.25 | 0.80 |

pixel-level reconstruction or extensive labeled data. It allows clinical use while preserving patient privacy. While JHU-VPT (JEPA) shows strong performance, further improvements are possible by increasing temporal resolution and incorporating finer motion features for feedback and skill assessment. Future work may explore domain adaptation to improve generalization across surgical environments.

## Acknowledgments

This research was supported by a grant from the National Institutes of Health, USA; R01EY033065. The content is solely the responsibility of the authors and does not necessarily represent the official views of the National Institutes of Health. Also, we would like to thank the Johns Hopkins Research IT team in IT@JH for their support and infrastructure resources, where some of these analyses were conducted, especially DISCOVERY HPC. Their commitment to advancing research has been invaluable in the successful completion of this study.

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

## Appendix A. More results of comparison on Pretraining dataset and Ablations

Table 5 presents a detailed comparison of JHU-VPT(JEPA) with several state-of-the-art methods for D99 Step Recognition under various data-regime settings (10%, 25%, 50%, and 100% of labeled data). The table categorizes methods by their pretraining datasets (e.g., Kinetics-400, D-450, D-2591) and the employed masking strategies (e.g., Random, Frame, Tube, Token Selection, and Multi-block).

Notably, JHU-VPT(JEPA), which is pretrained on the large and diverse D-2591 dataset using a Multi-block masking approach, achieves the highest performance in low-data regimes (62.2 at 10%, 65.86 at 25%, and 70.42 at 50%). These results underscore JHU-VPT(JEPA)'s ability to learn robust representations that are particularly effective when labeled data is scarce. While some methods, such as GLSFormer, surpass JHU-VPT(JEPA) at the full data regime (100%), our approach offers a compelling advantage in scenarios where extensive labeled data is unavailable.

Overall, these findings highlight the effectiveness of combining extensive pretraining with tailored masking strategies, positioning JHU-VPT(JEPA) as a strong candidate for applications with privacy constraints and limited annotation resources.

Figure 3 shows the step recognition accuracy across different dataset splits after full fine-tuning. Both our JHU-VPT(JEPA) and the D-2591 models (VideoMAE pretraining) consistently outperform CSMAE on the Cataract-1k and Cataract-1k-JHU datasets, reinforcing the robustness of our pretraining with a large dataset.

Table 5: Comparison of JHU-VPT(JEPA) with other state-of-the-art methods on D99 Step Recognition, under different data-regime settings and pretraining datasets.

| Methods | Pre-training Dataset | Masking | Data Regime (%) | | | |
|---|---|---|---|---|---|---|
| | | | 10 | 25 | 50 | 100 |
| MaskFeat (Wei et al., 2022) | Kinetics-400 | Random | 47.28 | 59.32 | 60.47 | 72.85 |
| GLSFormer (Shah et al., 2023) | Kinetics-400 | - | 47.19 | 61.54 | 63.76 | **80.24** |
| VideoMAE (Tong et al., 2022) | D-450 | Frame | 48.62 | 58.73 | 60.84 | 70.91 |
| STMAE (Feichtenhofer et al., 2022) | D-450 | Random | 52.37 | 60.42 | 63.58 | 74.16 |
| VideoMAE (Tong et al., 2022) | Kinetics-400 | Tube | 46.16 | 59.76 | 60.99 | 73.35 |
| VideoMAE (Tong et al., 2022) | D-450 | Random | 50.24 | 60.89 | 62.34 | 72.98 |
| VideoMAE (Tong et al., 2022) | D-450 | Tube | 52.11 | 61.59 | 63.72 | 74.39 |
| CSMAE (Shah et al., 2025a) | D-450 | Token Selection | 54.75 | 63.12 | 65.83 | 78.14 |
| JHU-VPT(JEPA) | D-2591 | Multi-block | **62.2** | **65.86** | **70.42** | 75.61 |

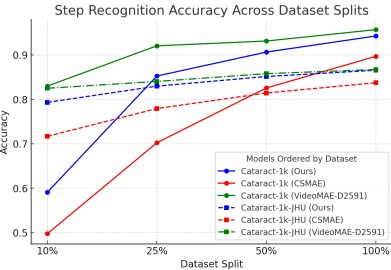

Figure 3: Step Recognition Accuracy across different dataset splits after complete fine-tuning.

For ablation experiments on model architecture, our results show that as the number of pretraining epochs increases, accuracy improves smoothly, demonstrating the scaling capabilities of the JEPA architecture. For instance, on D99 accuracy increased from 65.65% to 77.20%, and on Cataract-101 from 80.42% to 89.82% (see Table 6).

Table 6: Epoch Ablation on Step Recognition Accuracy. Accuracy values (in %) for each checkpoint are shown for the D99 and C101 datasets. The best performance for each dataset is highlighted in bold.

| Dataset | 60 | 100 | 200 | 240 | 300 |
|---|---|---|---|---|---|
| D99 | 65.65 | 69.97 | 70.84 | 74.03 | **77.20** |
| Cataract-101 | 80.42 | 83.72 | 86.38 | 88.17 | **89.82** |

Our ablation study evaluating various classification strategies—ranging from simple linear probing and attentive probing to full fine-tuning—shows that full fine-tuning generally achieves the best performance. However, attentive probing offers a compelling trade-off, particularly for the D99 dataset where it outperforms full fine-tuning, as it requires less computation and enables privacy-aware fine-tuning by avoiding back-propagation through the visual encoder. These findings suggest that the choice of classification strategy can be tailored based on labeled data availability, privacy requirements, and computational resources (see Table 7).

Table 7: Ablation experiments on JHU-VPT(JEPA) for the step recognition task with Full Finetuning, Attentive Probing, and Linear Probing across four datasets.

| Dataset | Full Finetuning | Attentive Probing | Linear Probing |
|---|---|---|---|
| Cataract-101 | **91.52** | 89.82 | 67.04 |
| Cataract-1k | **94.25** | 79.58 | 52.51 |
| Cataract-1k-JHU | **86.55** | 83.65 | 59.04 |
| D99 | 75.61 | **77.20** | 48.75 |

