# OpenReview forum: "A Vision Foundation Model for Cataract Surgery Using Joint-Embedding Predictive Architecture"
_MIDL.io/2025/Conference — MIDL 2025 Poster_

### Official Review · Reviewer_fdzR · 2025-02-10

**Confidence:** 3
**Preliminary Rating:** 4
**Recommendation:** Oral

**Summary:**

### **Key Concepts, Methodology, and Impact**

This paper presents **JHU-VPT(JEPA)**, a vision foundation model designed for **cataract surgery video analysis**, leveraging **Joint-Embedding Predictive Architecture (JEPA)** to learn **spatiotemporal representations** without requiring labeled data. Unlike conventional approaches such as **contrastive learning** or **masked autoencoders (MAE)**, which focus on pixel-level reconstruction, this model instead predicts **latent-space features**. This not only enhances generalization across diverse surgical environments but also ensures **privacy preservation**.

The model is pr-etrained on 2,591 surgical videos and evaluated across three critical tasks:
**step recognition, surgical feedback, and skill assessment**.
Experimental results demonstrate that JHU-VPT(JEPA) significantly outperforms **VideoMAE** and other state-of-the-art models.
Remarkably, it achieves **up to a 70% improvement in step recognition** even when trained on **only 10% of labeled data**, underscoring its **efficiency in low-data settings**.

A key component of this framework is the integration of **multi-block masking** and an **EMA-based target encoder**, which collectively prevent model collapse while ensuring the extraction of meaningful representations.
These findings underscore the potential for **scalable, privacy-conscious AI models in surgical education and real-time operative feedback**, paving the way for advancements in **patient care and surgical training**.

**Strengths:**

- JEPA for Surgical AI – Predicts latent-space features instead of pixels, reducing computation and enhancing privacy.
- Large Dataset – Trained on 2,591 cataract surgery videos for better generalization.
- High Efficiency – Achieves 35.12% step recognition accuracy with just 10% labeled data (vs. 20.70% for VideoMAE).
- Privacy-Preserving – Learns from feature embeddings, enabling secure data sharing & federated learning.
- Comprehensive Evaluation – Assessed on step recognition, feedback prediction, and skill assessment with rigorous methodology.

**Weaknesses:**

- Full Fine-Tuning Limitations – Underperforms against GLSFormer & CSMAE in full fine-tuning, questioning pretraining scalability.
- No Ablation Studies – Lacks analysis on JEPA’s advantage over MAE & contrastive learning.
- Missing Real-Time Evaluation – No inference speed assessment, crucial for surgical AI.
- Limited Dataset Justification – Unclear case diversity, camera angles, and demographic representation.
- No Public Code – Lack of pre-trained models & dataset access reduces reproducibility.

**Detailed Comments:**

### **Introduction**
- **Condense for clarity**: Break down long sentences for better readability.
- **JEPA Brief Intro**: Start with a short definition before introducing the full name. For example:
  *"JEPA, a self-supervised learning approach for feature representation, enables efficient learning by predicting missing data. The Joint-Embedding Predictive Architecture (JEPA) has shown promise in various applications..."*
- **Summary before technical depth**: Add a high-level overview before diving into complex math. Something like:
  *"At its core, JEPA minimizes feature prediction loss, which ensures robust latent space learning. Below, we define the key mathematical components driving this process."*

### **Methodology**
- **Clarify distinctions from prior models**: Clearly state what sets JHU-VPT(JEPA) apart. For example:
  *"Unlike traditional contrastive learning models, JHU-VPT(JEPA) focuses on multi-block masking strategies, allowing more efficient latent space learning in surgical video analysis."*
- **Visual representation**: A diagram showing multi-block masking would greatly improve clarity. Consider including a side-by-side comparison of JEPA vs. previous methods.

### **Experimental Design & Results**
- **Highlight key improvements**: In tables, bold the best results to make them stand out.
- **Model limitations**: Discuss scenarios where the model struggles, such as handling occlusions in surgery or variations in lighting conditions.

### **Future Work & Conclusion**
- **Higher frame rates & generalizability**: Suggest using higher FPS to capture fine-grained surgical motions and adapting JEPA to different types of surgeries.
- **Domain adaptation**: Discuss potential applications in broader medical imaging fields.
- **Ethical considerations**: Address privacy concerns, such as ensuring patient data security and mitigating bias in surgical AI models.

### **Final Touches**
- **Consistent citations**: Ensure uniform formatting across all references.
- **Bold key results**: Improve readability by making significant figures stand out.
- **Layman-friendly summary**: Consider adding a short section or appendix explaining JEPA's impact in simple terms for a broader audience.

**Justification Of The Preliminary Rating:**

The paper earns a high rating for its novel JEPA-based approach, strong technical foundation, and comprehensive experimentation:

Significance & Novelty – Introduces a privacy-preserving vision foundation model for cataract surgery, addressing dataset limitations and improving generalizability.

Technical Soundness – Well-justified latent feature prediction, a robust training strategy, and clear benchmarking against state-of-the-art methods.

Experimental Rigor – Demonstrates strong performance across multiple datasets and tasks, including step recognition, feedback, and skill assessment.

Clarity & Presentation – Well-structured but could benefit from simplifying dense sections and adding visuals for better clarity.

Impact & Future Potential – Highly applicable to surgical education and assessment; potential for expansion to other procedures through domain adaptation.

**Questions To Address In The Rebuttal:**

None

**Special Issue:**

Yes

---

> ### Author Response · Authors · 2025-03-08
>
> We thank the reviewer for their thorough evaluation and valuable comments.
> 1. Our primary objective is to learn robust and transferable representations for downstream tasks, as shown by strong results in linear and attentive probing experiments(Table 1), which underscore the quality and generalizability of the learned features.
> 2. In the revised manuscript, we have expanded our ablation studies to further highlight these advantages. i.e., Table 7 and Table 8
> 3. All evaluations using our pre-trained models were performed offline with ViT-16-B models, ensuring consistent inference times.
> 4. Additionally, we are willing to work to enable others to access our model checkpoints in the manner allowed by our institutional policies.
> We appreciate the constructive comments of the reviewer that have contributed to improving the manuscript.

---

### Official Review · Reviewer_syS1 · 2025-02-13

**Confidence:** 5
**Preliminary Rating:** 2
**Recommendation:** Poster

**Summary:**

The paper introduces a vision foundation model designed to assist cataract surgery by employing a joint-embedding predictive architecture. This model integrates preoperative imaging and intraoperative video data to provide real-time analytics and decision support during surgical procedures. The authors conducted experiments demonstrating the model's potential to enhance surgical precision and outcomes.

**Strengths:**

- The integration of preoperative and intraoperative data showcases a comprehensive approach, potentially leading to more accurate predictions and improved surgical outcomes.
- The emphasis on real-time analytics can significantly aid surgeons during procedures, enhancing precision and reducing the likelihood of complications.

**Weaknesses:**

- The architecture employed in the paper lacks novelty. The Joint Embedding Predictive Architecture (JEPA) has been previously applied in various domains, including medical imaging (eg. https://arxiv.org/html/2410.13867v1)
  - The paper does not provide a detailed explanation for the selection of the Joint Embedding Predictive Architecture (JEPA) model. Understanding the rationale behind choosing this specific model is crucial, as it would offer insights into how the architecture aligns with the objectives of enhancing cataract surgery through the integration of preoperative and intraoperative imaging
- The paper does not provide adequate details about the dataset used, such as its size, diversity, and specific features. This lack of information makes it difficult to assess the model's generalizability and applicability across different patient populations and surgical scenarios. eg. If the dataset lacks diversity in terms of patient demographics or disease variations, the model might develop biases, leading to less accurate predictions for underrepresented groups.
- The model may have been trained extensively on the available dataset, which could lead to overfitting. This means the model performs well on the training data but may not generalize effectively to new, unseen data.

**Detailed Comments:**

-

**Justification Of The Preliminary Rating:**

The paper lacks novelty in its use of the Joint Embedding Predictive Architecture (JEPA), a model previously applied in medical imaging, such as in self-supervised learning from ECG data. Additionally, the paper provides insufficient details about the dataset, including its size and diversity, making it difficult to assess the model's generalizability and potential biases.

**Questions To Address In The Rebuttal:**

Please look at the weakness section

**Special Issue:**

No

---

> ### Author Response · Authors · 2025-03-08
>
> *Comparison with JEPA for ECG data* : We thank the reviewer for sharing the reference on JEPA for ECG data. However, that work focuses on 2D representations of 12-lead ECG time-series signals, which fundamentally differ from our application that involves spatio-temporal cataract surgical videos. In ECG traces, where the data are images of waveforms or waveforms and the correlations within the time-series are limited and rhythmic. On the other hand, surgical videos capture the visual and procedural context of surgery, including instruments, wound details, and temporal coherence that is not necessarily rhythmic. In contrast, we also demonstrate the use of pretrained models for tasks such as step recognition, skill assessment, and surgical feedback, and we highlight their utility in low-data settings. We will update our manuscript to cite the referenced ECG paper for completeness.
>
> *Rationale for Choosing JEPA.* : We appreciate the reviewer’s feedback and have clarified the motivation and benefits of JEPA in the revised manuscript. First, our JEPA method masks tokens and predicts them in a latent feature space instead of reconstructing raw pixel values, reducing the impact of learning low-level artifacts such as reflections and blur. By focusing on higher-level semantic details, the model can predict more relevant representations for tasks like recognition, feedback, and skill assessment. Second, by not using a pixel decoder for reconstruction and avoiding separate reconstruction loss, it simplifies the learning objective and eliminates implicit dual-optimization of features and reconstruction concerns, focusing on meaningful features over pixel-perfect reconstructions. Finally, the use of an exponential moving average (EMA) stabilizes training and mitigates gradient noise, which is particularly beneficial in surgical video analysis where lighting variations and rapid movements can otherwise disrupt pre-training.
>
> *Dataset Composition:* : We appreciate the reviewer’s comment regarding dataset details. In our work, we assembled a multi-institutional dataset of 2,591 unique unlabeled cataract surgery videos. The dataset comprises 1,838 videos from one institution averaging 30 minutes at 59 fps and 753 videos from Cataract-1k (Ghamsarian et al., 2024) averaging 8 minutes, as summarized in Table 1, and Sec 2.7. Detailed patient demographics were not recorded for Cataract-1k in the published paper, and our internal dataset does not include patient- or surgeon-level identifiers; however, our dataset reflects the patient population seen at our institution. Future work will focus on expanding the pretraining data to incorporate sources from different geographical regions, further enhancing dataset and algorithm generalizability.
>
> Table 1: Statistics of the pretraining dataset.
>
> | Dataset            | \# of Videos | Average Duration | FPS  |
> |--------------------|--------------|------------------|------|
> | JHU-1838 (Internal) | 1,838        | 30 min           | 59   |
> | Cataract-1k        | 753          | 8 min            | 59   |
> | Total              | 2,591        | --               | 59   |
>
> *Overfitting concerns:* : We thank the reviewer for raising concern regarding overfitting. However, It is important to note that our pretraining dataset is **completely disjoint** from the evaluation sets, which include Cataract-101, D99, Cataract-1k, and Cataract-1k-JHU. As detailed in Sec.~2.7, we removed all videos with annotations—used for fine-tuning and testing—from the pretraining stage. Notably, the performance on Cataract-101 and D99, which come from entirely different sources compared to Cataract-1k and JHU-1838 (institutions and surgeons, respectively), serve as good tests for assessing the robustness of the learned feature extractor. These results demonstrate that extensive pretraining did not compromise the model's ability to generalize to new, unseen data.

---

> > ### Author Response · Authors · 2025-03-13
> >
> > We believe that we have thoroughly addressed the reviewer's concerns and kindly request the reviewer to let us of any additional issues following our response. We thank the reviewer for their time.

---

### Official Review · Reviewer_ARN9 · 2025-02-20

**Confidence:** 3
**Preliminary Rating:** 3
**Final Rating:** 4

**Summary:**

The authors introduce a JEPA-based approach for cataract videos. The model is pretrained on a large dataset.

**Strengths:**

The model was pre-trained on a large dataset, and the final quantitative results are promising, demonstrating the applicability of the proposed method. The model is capable of learning rich spatio-temporal representations.

**Weaknesses:**

The proposed paper lacks significance tests and qualitative evaluations. Additionally, it does not include an ablation study to assess each component's contribution to the final results. The methodology contribution of this paper is minimal.

**Detailed Comments:**

Figure 1 could be further improved, as the fonts are really small in the graph.

**Justification Of The Final Rating:**

Overall, the authors answered the majority of my questions well. The major concern of me for this paper still lays in the lack of novelty; however, due to this paper type is "Validation or Application". Thus, I choose to update my initial rating for this paper.

**Justification Of The Preliminary Rating:**

The results are promising but lack important testing. The contributions of this paper are minimal; however, since the focus is primarily on applications, the quantitative results are still meaningful.

**Questions To Address In The Rebuttal:**

1. There is no qualitative evaluation: how does the qualitative results compare to other competing methods?

**Special Issue:**

No

---

> ### Author Response · Authors · 2025-03-08
>
> 1. We thank the reviewer for their comment and have updated the Qualitative Results in the paper (as shown in Fig. 2). The visualization shows that our JHU-VPT (JEPA) model not only outperforms VideoMAE in overall quality but also exhibits improvements at step-transition points and reduces temporal jitter in step prediction.
> 2. Thanks for mentioning about significance test, our analyses show that the accuracy of our method is statistically significantly better than that with chance prediction (p-value < 0.001)
> 3. We have added ablation studies in the revised manuscript, where we examine the improvement in step recognition accuracy with an increasing number of pretraining epochs (Table 6) and evaluate different classification strategies (Table 7), ranging from simple linear probing to attentive probing and full fine-tuning. Our results show that as the number of pretraining epochs increases, accuracy improves smoothly, demonstrating the scaling capabilities of the JEPA architecture. Furthermore, our experiments indicate that the choice of classification strategy can be tailored based on labeled data availability, privacy requirements, and computational resources.
>
> Table 1: Epoch Ablation on Step Recognition Accuracy. Accuracy values (in %) for each checkpoint are shown for the D99 and C101 datasets. The best performance for each dataset is highlighted in bold.
>
> | Dataset       | 60    | 100   | 200   | 240   | 300   |
> |---------------|-------|-------|-------|-------|-------|
> | D99           | 65.65 | 69.97 | 70.84 | 74.03 | **77.20** |
> | Cataract-101  | 80.42 | 83.72 | 86.38 | 88.17 | **89.82** |
>
> Table 2: Ablation experiments on JHU-VPT(JEPA) for the step recognition task with Full Finetuning, Attentive Probing, and Linear Probing across four datasets.
>
> | Dataset        | Full Finetuning | Attentive Probing | Linear Probing |
> |----------------|-----------------|-------------------|----------------|
> | Cataract-101   | **91.52** | 89.82             | 67.04          |
> | Cataract-1k    | **94.25** | 79.58             | 52.51          |
> | Cataract-1k-JHU| **86.55** | 83.65             | 59.04          |
> | D99            | 75.61           | **77.20** | 48.75          |

---

> > ### Author Response · Authors · 2025-03-13
> >
> > We believe that we have thoroughly addressed the reviewer's concerns and kindly request the reviewer to let us of any additional issues following our response. We thank the reviewer for their time.

---

### Author Rebuttal · Authors · 2025-03-08

**Rebuttal:**

We appreciate R1 (ARN9), R2 (YSwy), and R3 (fdzR) for their comments and constructive feedback. We are encouraged that the reviewers recognize our **pretraining approach, built on a large dataset (R1, R3), as capable of learning rich spatiotemporal representations (R1) and delivering strong quantitative results (R1, R2, R3)**. We also appreciate the acknowledgment of our dataset integration for improved pretraining (R2) and the privacy-preserving yet efficient design of our method with rigorous evaluation across multiple tasks (R3).

In response to the feedback, we have added **qualitative evaluations (Fig.~2) on four datasets—Step Recognition, Cataract-101, D99, Cataract-1k, and Cataract-1k-JHU—showing improved predictions, especially at step-transition boundaries, and reduced temporal jitter compared to VideoMAE**. We have also included **ablation studies** to analyze the impact of increasing pretraining epochs on step recognition accuracy (Table 6) and to compare different classification strategies—linear probing, attentive probing, and full fine-tuning—showing the flexibility of our model based on labeled data availability, privacy needs, and computational resources (Table 7).

Additionally, we have clarified the rationale for choosing JEPA, explaining how predicting masked tokens in latent space and avoiding pixel-level decoders help learn better spatiotemporal representations in the Introduction Section of the paper. We have updated **Table 1 for comparison with VideoMAE on different probing experiments**, updating previous error in the attentive probing result and added **Table 3 showing model performance under low-data regime**. Grammatical errors have been revised throughout the paper for improved readability. We also validate our model’s significance performance, noting its significant difference from chance (p~$<$~0.001). To address overfitting concerns, we confirm that the pretraining dataset remains entirely separate from the fine-tuning and test sets. Dataset details (Sec. 2.7) have been expanded to reflect the **multi-institutional nature of our pretraining dataset**, consisting of 2,591 unique cataract surgery videos.

We again thank the reviewers for their valuable feedback, which has helped refine our work's clarity, fairness, and rigor. More detailed responses are provided in individual comments.

**Supporting Material:**

/attachment/6bb506584fb74dc45e6f701e0b452d249c2e54c6.pdf

---

### Meta-Review · Area_Chair_bsV6 · 2025-03-22

**Recommendation:** Accept (Poster)
**Confidence:** 4

**Metareview:**

This paper received 2 weak accept and 1 weak reject. Based on the reviews and authors' careful responses, I recommend accept.